# From Clutter to Clarity: Visual Recognition through Foveated Object-Centric Learning (FocL)

**Amitangshu Mukherjee    Deepak Ravikumar    Kaushik Roy**
Elmore Family School of Electrical and Computer Engineering
Purdue University, West Lafayette, IN 47906
{mukher44, dravikum, kaushik}@purdue.edu

## Abstract

Humans perceive the world through active vision, using rapid eye movements to focus on task relevant regions while ignoring irrelevant background clutter. Inspired by this, we introduce FocL (Foveated Object Centric Learning), a training strategy that biases image classification models toward label consistent object regions by replacing full images with foveated crops. Standard training encourages models to rely on spurious context, which degrades generalization and increases memorization, especially for hard examples in the tail of the sample difficulty distribution. FocL simulates saccades by (1) jittering fixation points around the annotated object and (2) extracting cropped regions centered on these points as foveated glimpses. This input restructuring reduces non foreground contamination, lowers mean training loss, accelerates convergence, and shifts hard samples closer to the center of the difficulty curve. In our analysis, FocL improves generalization by up to 15% on oracle crops and improves out-of-distribution generalization from ImageNetV1 to V2 by over 7pp when paired with modern segmentation models like SAM. This reduced reliance on spurious correlations increases the mean PGD L2 adversarial distance required to flip a training set prediction by 61% and directly resolves learning difficulty for the top 1% memorized samples in ImageNet, reducing their cumulative sample loss by 62.5%. By training on foveated crops, FocL requires 56% less data to exceed the performance of standard models. FocL thus offers a simple path to more robust, and reliable visual recognition.

## 1 Introduction

Deep neural networks often achieve high performance by relying on spurious correlations between labels and irrelevant background features Bayat et al. [2025], Geirhos et al. [2020], rather than learning robust object-centric representations. This hinders generalization on hard examples in the tail of the sample-level difficulty distribution, even when class frequencies are balanced Arpit et al. [2017], Usynin et al. [2024]. An example of sample-level difficulty is the sample's training loss (or its gradient norm), which quantifies how challenging it is to learn. The left sub-panel of Figure 1 illustrates this: harder examples concentrate in the tail under difficulty measures (e.g., loss or curvature) Garg et al. [2024], Ravikumar et al. [2024, 2025]. These instances often lead to memorization, where models overfit to background context, dataset artifacts, or unrelated co-occurring objects instead of focusing on the labeled foreground object Brown et al. [2021], Feldman and Zhang [2020]. The right sub-panel of Figure 1 illustrates common failure sources. These failures include unlabeled distractors like humans, and label ambiguity from multiple objects in a single annotated image, for example, a "Labrador" image that also contains other dog breeds.

Many methods attempt to address learning in such long tail settings Kang et al. [2020], Ren et al. [2020], Tan et al. [2021], Yun et al. [2019], Zhang et al. [2018, 2023], however, they still train on

39th Conference on Neural Information Processing Systems (NeurIPS 2025) Workshop: Reliable ML from Unreliable Data.

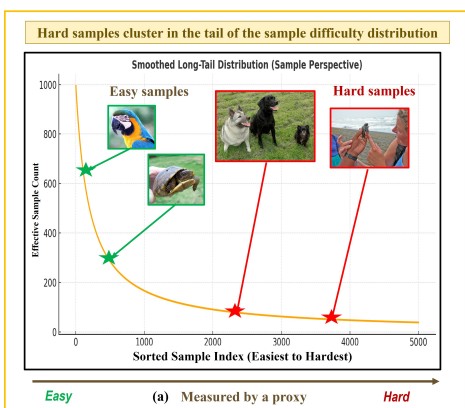 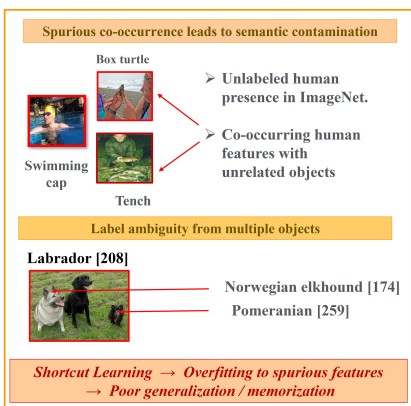

Figure 1: Figure illustrates key challenges that drive memorization and hinder generalization in visual recognition. (Left) A conceptual long-tail curve of sample-level difficulty, with harder examples concentrated in the tail and difficulty measured via proxies such as sample loss. (Right) Two major sources of sample-level hardness: (a) Spurious correlations from unlabeled co-occurring entities (e.g., humans) cause models to overfit to background context; (b) Label ambiguity from multi-object images (e.g., a "Labrador" sample also containing other dog breeds) introduces confusion. These effects weaken object-label consistency and promote reliance on shortcuts.

full, cluttered images and hence, hard examples remain hard. In contrast, we target sample level difficulty: the individual examples that challenge a model even when classes are balanced. This leads to a natural question: can we improve generalization by presenting object centric, foveated views analogous to how humans focus on the most informative regions, thereby filtering out irrelevant and spurious features?

To explore this, we draw on insights from biological vision. As illustrated in Figure 2(a), human perception operates through an active vision system that combines goal-directed sampling with object-centric encoding. The initial visual input is captured via peripheral vision, which provides coarse information across the scene. Based on this, saccadic eye movements shift the fovea, the high-acuity center of the retina toward salient targets. According to the two-stream hypothesis Clark [2013], Goodale and Milner [2004], Milner and Goodale [1992], Mishkin et al. [1983], Sakuraba et al. [2012], Ungerleider and Haxby [1994], the dorsal stream computes where to look by identifying spatially informative regions. In parallel, the ventral stream processes Eckstein [2011], Shao et al. [2024] the high-resolution foveated input to determine what is being observed, extracting semantic features such as shape and identity. This foveated mechanism allows humans to extract consistent, object-centered representations across varied contexts, forming the basis for robust generalization.

Inspired by these principles of biological vision, we introduce FocL, which trains networks on foveated object-centric crops that isolate the foreground and thereby simplify learning and boost generalization. We emulate saccades by jittering bounding-box centers to generate multiple, object-focused glimpses. By suppressing background clutter and isolating task-relevant regions, *FocL reduces sample complexity, shifting hard instances from the tail toward the mode of the distribution.* Rather than requiring models to learn from visually complex scenes, FocL restructures the input space to emphasize object–label consistency, effectively reframing image classification as a simpler, more targeted task. FocL's object-centric strategy improves generalization on foveated inputs and reduces memorization. FocL models require larger adversarial perturbations and convergence faster, enabling learning from less data. Figure 2 visualizes this effect in a t-SNE projection, where FocL produces tighter semantic clusters. In contrast, a standard model entangles distinct classes, such as *hippopotamus* vs. *water buffalo*, by relying on their shared background context (e.g., water). By focusing on foveated object regions, FocL learns cleaner semantic boundaries and more robust representations.

**Our contributions are as follows:**

- **FocL: Object-Centric Training Strategy.** We introduce **FocL**, a training method that generates object-centric glimpses by jittering ground-truth boxes, focusing models on foreground features.
- **Improved Generalization on Object Features.** FocL boosts Top-1 accuracy by ~15% on oracle object crops where standard models fail. Following the dorsal-ventral stream

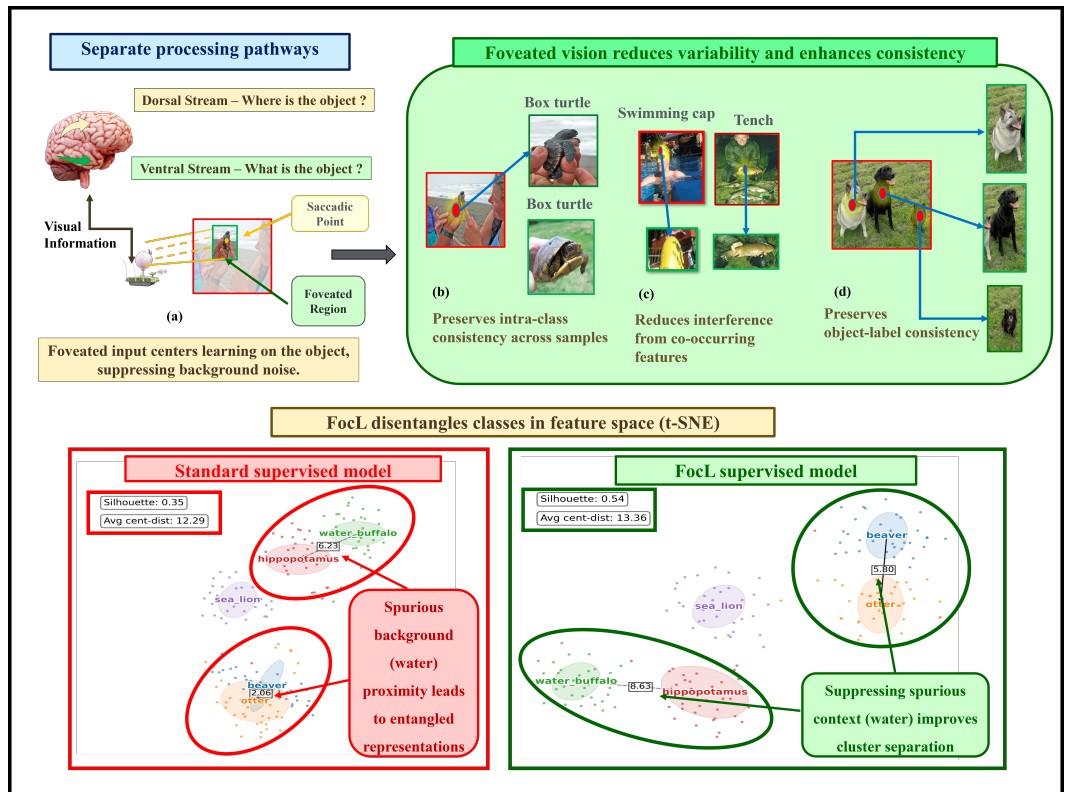

Figure 2: **FocL emulates human foveated vision to improve generalization by suppressing spurious context.** (a) FocL uses object-centric glimpses inspired by human visual streams to focus on relevant features. (b–d) This object-centric bias leads to more robust learning outcomes (intra-class consistency, reduced interference, object-label alignment). (Bottom) *t*-SNE: FocL (right) achieves better class separation (silhouette +0.19, avg. centroid dist. +1.07), unlike standard models (left) that confuse distinct classes (e.g., hippopotamus/water buffalo) by relying on shared water backgrounds.

hypothesis, we demonstrate that a FocL-trained classifier (the "what" stream) excels when paired with a powerful external proposal model (the "where" stream), improving out-of-distribution accuracy by over 7pp when using SAM for localization.

- **Reduced Memorization and Spurious Correlation.** FocL mitigates memorization of non-robust features: (i) it increases the $\ell_2$ adversarial distance to flip training sample predictions by 61% (evaluated on respective inputs) and (ii) it directly addresses the hardest examples, confirmed by a 62.5% reduction in learning difficulty for the top 1% of memorized samples.

- **Enhanced Learning Dynamics and Data Efficiency.** Improved focus and reduced learning difficulty translate to smoother optimization (46% lower mean gradient norm) and enable FocL to match or exceed baseline performance with 56% less training data.

## 2 Related Work

We provide here a compact yet comprehensive survey of work most relevant to FocL; an expanded version is provided Supplementary.

**Object-centric and foreground-focused learning.** Unsupervised methods such as MONet Burgess et al. [2019] and Slot Attention Locatello et al. [2020] aim to disentangle objects, whereas attention add-ons (e.g., CBAM Woo et al. [2018]) and discovery pipelines like CutLER Wang et al. [2023] modulate full-image features or mask foregrounds after the fact. A related thread learns *where* to look through iterative policies, exemplified by RANet Mnih et al. [2014], Saccader-style models Elsayed et al. [2019], GFNet Wang et al. [2020], FABLE Ibrayev et al. [2024a], and FALcon Ibrayev et al. [2024b]. **FocL instead feeds the network only supervised foveated crops**, hard-coding foreground–label consistency and suppressing background interference; object localization can be

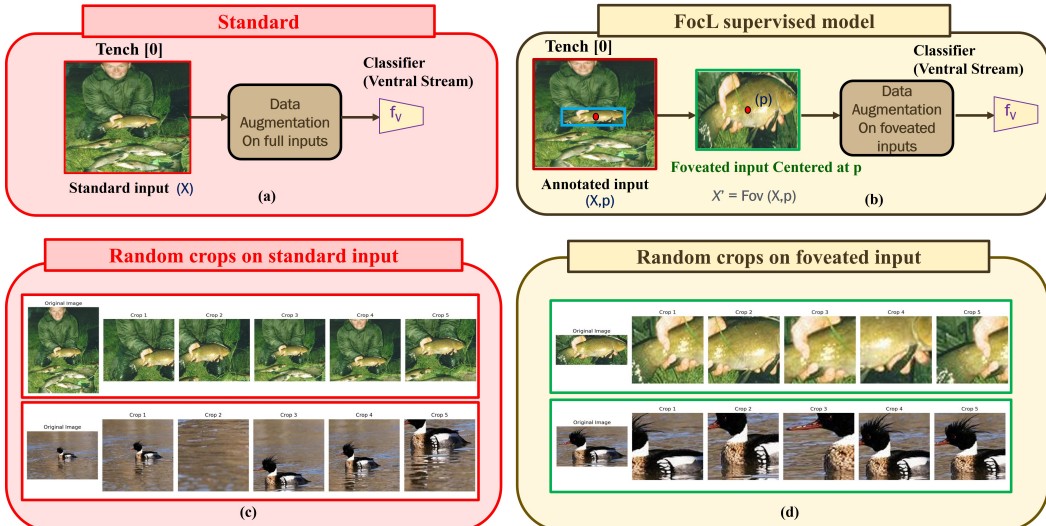

Figure 3: (a) Standard training uses the full image. (b) FocL replaces the raw input with foveated crop/crops centered on the annotated object. (c–d) Effect of Random-Resized-Crop augmentation under both pipelines. Each row shows the original image (left) followed by five crops seen across training epochs. In (c), full-image augmentation often captures irrelevant background (e.g., a fisherman's jacket or just water), encouraging spurious correlations. In contrast, (d) applies the same augmentations to foveated crops, yielding object-centric views that preserve foreground features. These cleaner views lead to more disentangled, object-aligned representations (see t-SNE, Figure 2).

delegated to external detectors at inference, while our focus is on the learning benefits of object-first bias.

**Memorization in long-tailed learning.** Networks typically fit frequent patterns before memorising rare, noisy, or atypical tail instances Arpit et al. [2017], Feldman and Zhang [2020]. Theory and evidence suggest such memorization can be necessary for accuracy under skewed data Brown et al. [2021], Usynin et al. [2024], yet it raises fairness, robustness, and privacy concerns Li et al. [2025]. Recent analyses propose proxies like Cumulative Sample Loss (CSL) Ravikumar et al. [2025] and link high input-loss curvature to memorised long-tail samples Garg et al. [2024], Ravikumar et al. [2024]. Unintended "déjà-vu" memorization has also been observed in SSL models Meehan et al. [2023], Kokhlikyan et al. [2024] and VLMs Jayaraman et al. [2024]. Rather than relying on models to navigate these complex memorization dynamics for hard samples, FocL restructures inputs to remove background clutter, simplifying hard examples and reducing reliance on brittle shortcut cues Geirhos et al. [2020]; unlike Mixup, CutMix, or logit adjustment Zhang et al. [2018], Yun et al. [2019], it tackles instance-level difficulty directly.

**Foveation, robustness, and our contribution.** Recent robustness-oriented work blurs or down-samples the periphery such as R-Blur for adversarial defence Shah et al. [2023], textural encodings for IID gains Gant et al. [2021], and active-vision systems that integrate multiple glimpses against transferable attacks Mukherjee et al. [2025]. These methods still retain background pixels, and the robustness–memorization relationship remains delicate; e.g., adversarial training can induce robust overfitting Dong et al. [2022]. FocL adopts a different stance: an *extreme cut-off* that entirely excises background via supervised crops. The observed increase in mean adversarial distance and the significant drop in CSL are beneficial by-products of FocL simplifying each learning instance, rather than outcomes of explicit robustness optimization. This improved learnability naturally leads to smoother convergence and more stable, generalizable representations.

## 3  Methodology

In this section, we introduce **FocL**, our multi-glimpse foveated learning framework for visual recognition. We begin by briefly reviewing standard supervised learning and highlight how data augmentation behaves differently when applied to global versus foveated inputs. We then describe the FocL in detail.

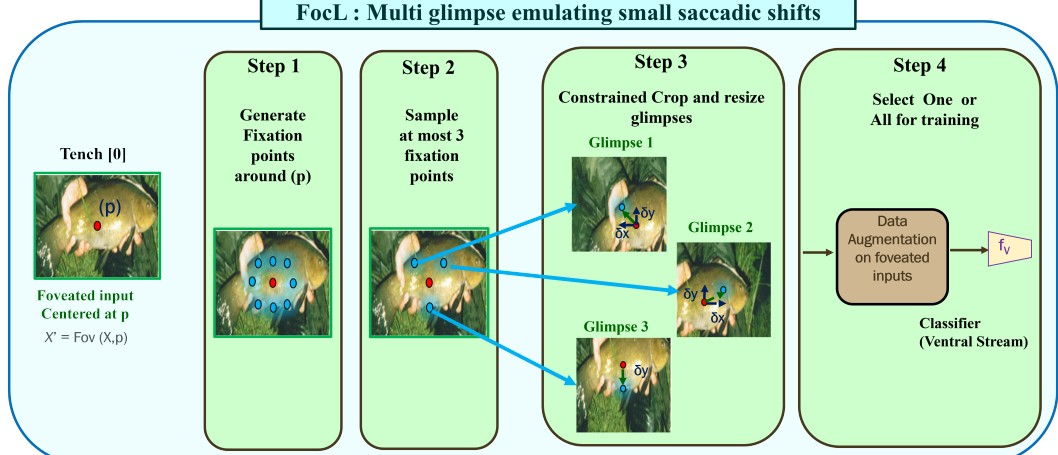

Figure 4: **FocL** with structured glimpse variation (Steps 1–4) simulates small saccadic shifts by jittering the fixation point and extracting up to three constrained crops around the object. Each glimpse is resized to the input resolution and used individually or jointly during training. These object-centric views reduce the influence of background clutter and encourage the network to focus on label-relevant foreground features, promoting stronger alignment between object structure and class semantics.

**Standard Supervised Training :** In conventional supervised pipelines (red panel, Figure 3a), the classifier $f$ is trained directly on full-resolution images using the standard cross-entropy loss:

$$\mathcal{L}_{\text{sup}} = \mathbb{E}_{(x,y)\sim\mathcal{D}}\left[\ell(f(x),\, y)\right],$$

where $(x, y)$ denotes an image-label pair and $\ell$ is the classification objective. Since the entire image $x$ serves as input, data augmentations such as random resized cropping, horizontal flipping, and color jitter are applied globally across both foreground and background regions. This global augmentation strategy can introduce semantic drift: the network may passively learn background features that are unrelated to the object label. As shown in Figure 3c, random crops observed during training emphasize irrelevant context, such as the fisherman's jacket instead of the tench (top row), or mostly water instead of the red-breasted merganser (bottom row). Such misaligned augmentations promote spurious correlations between background and label, causing the model to overfit to incidental context rather than learning object-centric, generalizable representations.

### 3.1   FocL: Foveated Object-Centric Learning

Given a labeled image $(x, y)$, we define the annotated bounding box as $b = (x_{\min}, y_{\min}, x_{\max}, y_{\max})$, and let its geometric center define a surrogate saccadic fixation point $p \in \mathbb{R}^2$. Since the label $y$ corresponds to the object within the box, the fixation is supervised and object-aligned. While biological foveation involves gradual spatial falloff and peripheral blur, we approximate it using a hard foveated glimpse by cropping around $p$ to retain the labeled foreground and discard most surrounding context. Since bounding boxes typically include some peripheral pixels, the glimpse may contain limited background; however, it remains substantially more object-aligned than full-image crops. This setup is visualized in Figure 3b.

Using this formulation, we instantiate **FocL**, a strategy that generates multiple object-focused glimpses (up to three per image) by applying small, controlled spatial and scale jitter around the initial fixation point $p$. These jittered glimpses serve to relax tight bounding boxes, emulate human-like saccadic sequences, introduce mild viewpoint variations, and mitigate geometric distortions from resizing. By primarily exposing the model to these varied object-centric views, **FocL** encourages a strong inductive bias towards foreground features over background clutter. Consequently, even when standard augmentations are applied, these glimpses maintain semantic consistency and preserve object identity (yellow panel, Fig. 3d).

For each image, we extract up to $k$ (tunable parameter) foveated glimpses and treat them as individual training examples sharing the same label. During training, these glimpses are included in the same mini-batch (i.e., not shuffled across images), enabling the model to jointly process multiple views

Table 1: Top-1 and Top-5 accuracy (%) on 773 held-out samples from ImageNet train. FocL (multi-crop) improves under oracle bounding box inference, while the standard model suffers a performance drop. Numbers in parentheses indicate absolute differences (bbox – full image) in percentage points.

| Model | Full Image | | BBox Inference | |
|---|---|---|---|---|
| | Top-1 | Top-5 | Top-1 | Top-5 |
| Standard | 65.33 | 87.27 | 60.28 (–5.05) | 82.62 (–4.65) |
| FocL | 58.64 | 80.59 | **75.79** (+17.15) | **94.07** (+13.48) |

Table 2: Comparison of full-image Top-1 accuracy with multi-glimpse inference using FALcon-style glimpses on 2K ImageNet validation samples. Numbers in parentheses indicate absolute differences in percentage points between full-image accuracy and each multi-glimpse metric.

| Model | Full Image | Avg | Voting | Voting Weighted |
|---|---|---|---|---|
| Standard | 63.23 | 61.25 (–1.98) | 60.10 (–3.13) | 60.52 (–2.71) |
| FocL | 53.27 | 61.45 (+8.18) | 60.68 (+7.41) | 61.37 (+8.10) |

of the same object and learn stable foreground–label mappings. The total loss is computed over all glimpses in the batch:

$$\mathcal{L}_{\mathrm{FocL}} = \mathbb{E}_{(x,y,p)\sim\mathcal{D}} \left[ \sum_{i=1}^{k} \ell\big(f(\mathrm{Fov}_i(x,p)), y\big) \right].$$

Here, $\mathrm{Fov}_i(x,p)$ denotes the $i^{\mathrm{th}}$ foveated crop generated around a distinct jittered fixation point $p_i$, sampled from a neighborhood of the base annotated center $p$. While each crop uses its own offset $p_i$, we denote it $\mathrm{Fov}_i(x,p)$ to indicate all glimpses are relative to the original $p$. The specific procedure for generating these valid, object-focused glimpses (illustrated in Figure 4) involves parafoveal sampling of candidate centers, selection based on image boundaries to maintain alignment, and distortion-aware cropping techniques. This design ensures robust learning primarily from foreground features under mild variations in position and scale. For a detailed algorithm, including specific jitter parameters and selection criteria, please refer to Supplementary.

## 4   Experiments

We evaluate FocL across three dimensions: **generalization under foveated inputs**, **robustness to memorization**, and **training efficiency**.

### 4.1   Does FocL improve generalization under foveated inputs?

**Experimental Setup**   We use the subset of ImageNet Deng et al. [2009], Russakovsky et al. [2015] with bounding box annotations (482K images). Models are standard ResNet-50 He et al. [2015]; the FocL model uses three foveated glimpses (Sec. 3). Both use identical standard augmentations (random resized crop, flip, color jitter), excluding advanced augmentations like RandAugment to ensure a controlled comparison. We evaluate generalization under three conditions: (1) oracle bounding box testing on a held-out set, (2) multi-glimpse aggregation with FALcon Ibrayev et al. [2024b], and (3) an upper-bound analysis with SAM Kirillov et al. [2023], Ravi et al. [2024]. The FALcon and SAM (V1) evaluations use the same 2,000 ImageNet validation samples for direct comparison, while the SAM (V2) OOD evaluation uses 2,000 random samples from the ImageNet-V2 (MatchedFrequency) set. Further details are in the supplementary material.

**Oracle Bounding Box Evaluation**   We first evaluate performance under ideal foveation using oracle bounding box crops (Table 1). This test reveals the standard model's reliance on spurious context, as its Top-1 accuracy drops by 5.05 pp on these inputs. In contrast, FocL thrives when background clutter is minimized, improving its accuracy by a significant 17.15 pp on the same object-centric crops.

Table 3: Comparison of "Any" correct Top-1 accuracy (%) using SAM-generated crops on 2,000 samples from ImageNet-V1 and ImageNet-V2. "Any" accuracy denotes whether *any* crop produced by SAM yields the correct Top-1 prediction.

| Model System | ImageNet V1 (%) | ImageNet V2 (%) |
|---|---|---|
| SAM + Standard | 78.63 ± 1.06 | 66.43 ± 1.04 |
| SAM + FocL | **83.13 ± 0.19** | **73.48 ± 0.75** |

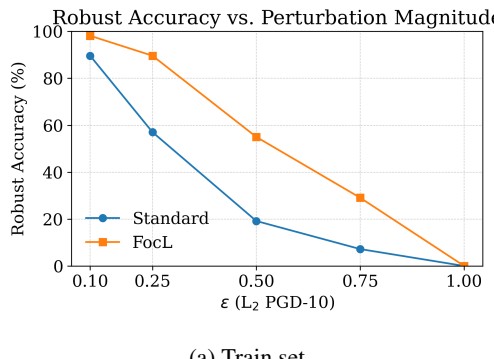

(a) Train set

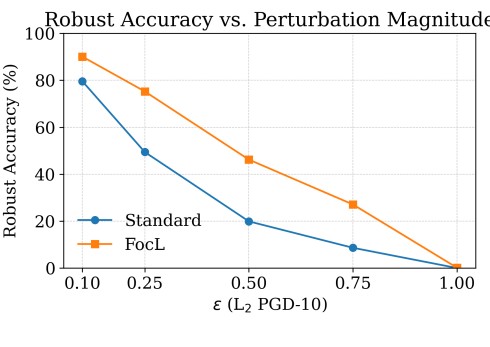

(b) Validation set

Figure 5: Robust accuracy of standard and FocL models against $\ell_2$ PGD-10 perturbations. At $\epsilon = 0.25$, FocL achieves 89.59% on the training set versus 57.05% for the standard model, a gap of 32.54 percentage points. On the validation set, the gap is 25.82 percentage points (75.24% vs. 49.42%). The standard model's accuracy drops sharply at small perturbation levels ($\epsilon < 0.25$), with many predictions flipped by minimal adversarial budgets. This behavior, especially evident on the training set, suggests memorization and a reliance on brittle and non-generalizable features. The degradation is less pronounced in FocL, likely due to its foveated training strategy which encourages robust object-centric feature learning by reducing background clutter.

**Inference within a Dorsal-Ventral Structure**    Evaluating a FocL-trained classifier (the ventral "what" stream) implicitly requires a dorsal-ventral structure. We measure classification performance given object-centric views from external dorsal ("where") models. We test this with two distinct dorsal stream types. First, we use FALcon Ibrayev et al. [2024b], an active vision framework, to assess performance under **multi-glimpse aggregation**. The standard model's accuracy declines under robust aggregation (e.g., Voting Weighted drops 2.71 points), revealing brittleness to varied object views. FocL, however, improves across all metrics (e.g., +8.10 points), demonstrating reliable generalization (Table 2). Second, we use the powerful foundation model SAM Kirillov et al. [2023], Ravi et al. [2024] to evaluate the classifier's **upper-bound performance** when provided with crops from a state-of-the-art segmenter. On ImageNet V1, the SAM+FocL system achieves a 4.5 pp higher "Any" correct accuracy. This advantage grows on the ImageNetV2 dataset Recht et al. [2019], where the FocL pipeline outperforms by over 7 pp (Table 3). While a monolithic standard classifier learns the spurious statistics of the ImageNetV1 distribution, our FocL model learns the intrinsic properties of the foreground objects (crucial for classification), making it more stable and confident when faced with the natural variations found in real-world scenarios.

## 4.2   Does FocL reduce memorization?

We answer this with two complementary analyses. First, we measure adversarial resistance to probe feature stability and robustness. Second, we analyze Cumulative Sample Loss Ravikumar et al. [2025], focusing on a cohort of verifiably memorized samples identified by connecting pre-computed FZ Feldman and Zhang [2020] scores to our training set's ImageNet indices.

**Adversarial Resistance**    To probe the stability of learned representations, we evaluate model robustness against PGD-$\ell_2$ adversarial attacks Madry et al. [2018] on a balanced ImageNet subset. For a fair comparison, models are attacked on their respective input types (full images vs. object crops). The results show that FocL learns significantly more robust representations. First, FocL requires a

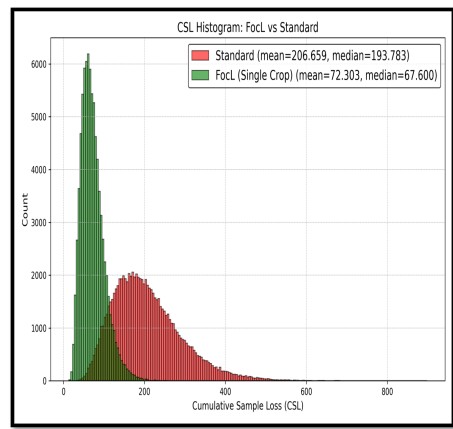 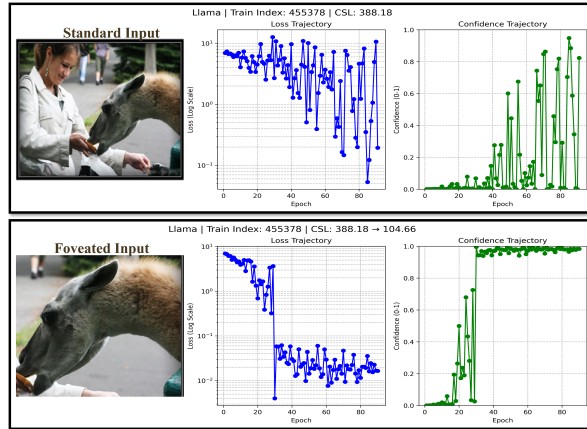

| (a) Distribution over entire training set | (b) Loss and confidence trajectories for a sample |
|---|---|

Figure 6: Cumulative sample loss (CSL) memorization proxy analysis. **Left:** Shift from tail to mode. FocL exhibits significantly lower mean and median CSL, and the distribution is tightly concentrated toward lower values, indicating that samples become easier to learn due to object-centric inputs and reduced contextual interference. **Right:** Example of a high-CSL sample from the Llama class. In the standard model, background elements like the human introduce semantic contamination, leading to noisy loss and confidence trajectories. With FocL, foveated input enables more stable learning, reflected in the smoother trajectories and a large CSL drop from 388.18 to 104.66.

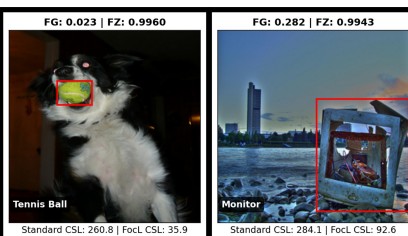 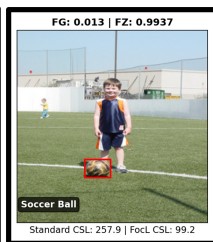 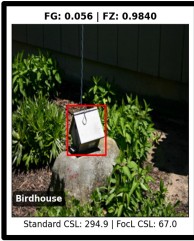 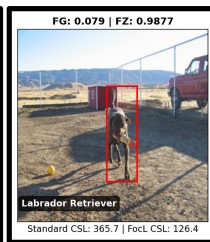

Figure 7: **Visualization of the top 1% of memorized ImageNet samples, identified by Feldman & Zhnag memorization scores**. These challenging examples feature small foreground objects surrounded by distracting context (e.g., FG of 0.013 for 'Soccer Ball'). By isolating the object, FocL drastically reduces the learning difficulty, evidenced by the large drop in **CSL** for each sample.

**61% greater** mean $\ell_2$ adversarial perturbation to flip a training set prediction ($\bar{d} = 0.6169$ vs. 0.3806 for the standard model), indicating more stable, harder-to-disrupt features. Second, FocL consistently maintains higher accuracy under increasing attack strengths ($\epsilon$), as shown in Figure 5. The standard model's sharp performance decline suggests a reliance on brittle, non-generalizable features often associated with memorization Carlini et al. [2019].

**Learning Difficulty and Memorization**   To investigate how FocL mitigates memorization, we analyze Cumulative Sample Loss (CSL) Ravikumar et al. [2025], a proxy for learning difficulty where higher values indicate harder-to-learn, often memorized, samples. On an aggregate level, FocL training dramatically reshapes the entire CSL distribution (Figure 6, left), reducing the mean CSL from 206.66 to 72.30. To understand the source of this improvement, we performed a targeted analysis using pre-computed memorization scores from Feldman & Zhang Feldman and Zhang [2020]. By intersecting the indices of the top 1% most memorized ImageNet samples with our 85K training set, we identified a cohort of 820 verifiably memorized samples. For this specific group, FocL was exceptionally effective, making **99.88%** of these hard samples easier to learn ($p < 0.001$). A dominant characteristic of this cohort is high contextual complexity, often from background clutter. We quantify this with the foreground-to-image area ratio (FG), defined as the bounding box area divided by the total image area. The mean FG for this cohort is just 0.457. This analysis empirically demonstrates that FocL's benefits stem from its ability to resolve learning difficulty for the most problematic samples, explaining the aggregate trend of shifting hard samples from the tail of the difficulty distribution toward the mode.

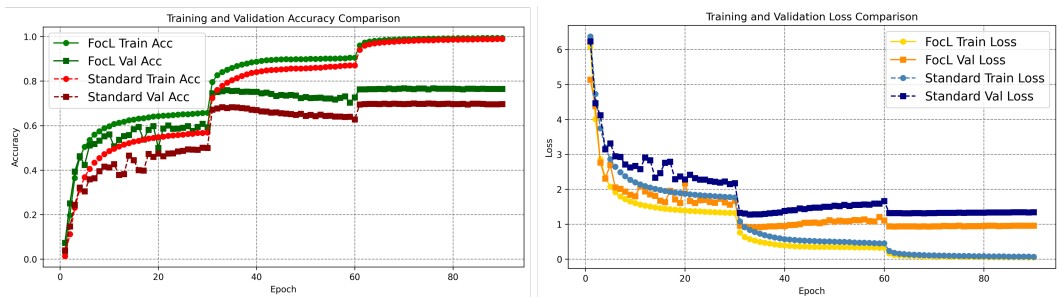

Figure 8: Training and validation accuracy and loss curves for FocL and standard models over 90 epochs. FocL converges faster and achieves lower training and validation loss throughout.

Table 4: Top-1 accuracy evaluated under Oracle bounding boxes. Corresponding number of training samples shown in brackets.

|  | Standard (452,954) | Standard (1,030,000) | FocL (452,954) |
| --- | --- | --- | --- |
| Top-1 Accuracy (%) | 60.28 | 63.13 | 74.51 |

### 4.3    Does FocL enhance learning efficiency and stability?

To enable fair comparisons, experiments in this section use the FocL single-crop variant against standard full-image training. We assess FocL's impact on training dynamics, optimization stability, and data efficiency.

**Training Dynamics and Convergence.** Learning curves on the 482K ImageNet subset (Figure 8) demonstrate FocL's superior training dynamics. FocL converges faster and consistently achieves lower training/validation losses and higher accuracies throughout epochs. This indicates that foveated inputs simplify the learning task for more stable and sample-efficient optimization.

**Smoother Optimization via Gradient Norm Reduction.** Analysis of $\ell_2$ gradient norms during training on the 85K subset further reveals FocL's stabilizing effect. FocL exhibited consistently lower gradient magnitudes. Specifically, the mean gradient norm per parameter (normalized by model size) was reduced by approximately 45.8% (from $1.49 \times 10^{-3}$ for standard to $8.08 \times 10^{-4}$ for FocL). This substantial drop suggests FocL creates a simpler optimization landscape with less gradient noise. (Absolute mean gradient norms: Standard $3.81 \times 10^{4}$, FocL $2.07 \times 10^{4}$).

**Data-Efficient Learning.** FocL's simplified learning paradigm translates to significant data efficiency (Top-1 accuracy, single crop evaluation, Table 4). FocL trained on 453K annotated images achieved 74.51% Top-1 accuracy when evaluated using oracle bounding boxes on the 773-sample held-out set. This substantially outperforms the standard model's 63.13% Top-1 accuracy obtained using 1.03M images (over twice the data). This underscores FocL's capability for more sample-efficient learning by fostering robust object-centric representations. Additional results in Supplementary.

## 5    Conclusion

FocL introduces a multi-glimpse training strategy that encourages models to learn object-centric features by reducing background clutter. This approach improves representation quality by mitigating spurious correlations, which disproportionately affects hard-to-learn samples. Our experiments demonstrate that FocL improves generalization, boosting accuracy by approximately 15% on oracle object crops and by over 7pp on out-of-distribution data (ImageNetV2) when paired with a modern segmentation model like SAM. We provide definitive evidence that FocL reduces memorization by targeting the most problematic examples; for the top 1% of memorized ImageNet samples, FocL makes 99.88% of them easier to learn. This enhanced feature learning results in more robust representations, evidenced by a 61% increase in the adversarial perturbation required to flip predictions. On the efficiency front, FocL leads to smoother convergence and achieves competitive accuracy with 56% less training data. Together, these results demonstrate that foveated training offers a simple and effective path to more robust, reliable, and data-efficient visual recognition.

# 6 Acknowledgments

This work was supported in part by, the Center for the Co-Design of Cognitive Systems (CoCoSys), a DARPA-sponsored JUMP 2.0 center, the Semiconductor Research Corporation (SRC), and the National Science Foundation.

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

## 7 Supplementary

## Expanded Related Work

This section expands on the related work most relevant to FocL, organized into three main areas: (1) object-centric and foreground-focused learning, (2) memorization and generalization in long-tailed settings, and (3) foveation-based methods for robustness and efficient learning.

### Object-Centric and Foreground-Focused Learning

Unsupervised object-centric models such as MONet Burgess et al. [2019] and Slot Attention Locatello et al. [2020] aim to decompose scenes into discrete object representations, but often struggle on complex natural images. Attention modules like CBAM Woo et al. [2018] reweight spatial and channel-wise features post hoc, while pipelines like CutLER Wang et al. [2023] attempt to discover and mask foregrounds, still operating over full-image inputs. A related class of models learns dynamic visual attention through iterative glimpses. RANet Mnih et al. [2014] uses a recurrent attention network to focus on different image regions over time, while Saccader Elsayed et al. [2019] and GFNet Wang et al. [2020] emulate saccadic movements and process glimpses within computational budgets. FABLE Ibrayev et al. [2024a] models a dorsal-ventral system using reinforcement learning to locate objects, and FALcon Ibrayev et al. [2024b] further introduces saccades and foveation, enabling active multi-object detection even from single-object training. These models mimic human

vision by sequentially sampling high-resolution glimpses, discarding background via task-adaptive attention. **FocL adopts a different paradigm. Rather than learning fixation policies, it uses supervised bounding boxes to directly crop foreground objects, fully removing background prior to training. This object-label alignment reduces contextual bias and simplifies training, focusing on the impact of this transformation on generalization, memorization, and convergence.**

### Memorization in Long-Tailed Learning

Deep networks tend to memorize rare, noisy, or atypical examples after first fitting frequent and simpler patterns Arpit et al. [2017], Feldman and Zhang [2020]. Arpit et al. Arpit et al. [2017] show that during training, networks prioritize learning generalizable patterns but eventually begin memorizing outliers and noisy data. Feldman and Zhang Feldman and Zhang [2020] further argue that memorization is not just incidental but sometimes essential for accurate predictions on tail samples, especially when such examples are underrepresented or conflict with dominant patterns in the data. Building on this, Brown et al. Brown et al. [2021] provide theoretical insights into why high-accuracy learners may be forced to memorize substantial information about training data in natural, long-tailed settings. Usynin et al. Usynin et al. [2024] offer a comprehensive survey of memorization across multiple regimes, categorizing its benefits and drawbacks with respect to generalization and privacy. Li et al. Li et al. [2025] take a systems-level view, framing memorization as central to the trustworthiness of machine learning systems. They explore its role across fairness, robustness, and data privacy, and propose a taxonomy to reason about these interactions based on data granularity such as class imbalance, noise, and atypicality. To characterize memorization quantitatively, Ravikumar et al. Ravikumar et al. [2025] introduce the Cumulative Sample Loss (CSL), which tracks the cumulative training loss per sample. They show that hard-to-learn and noisy samples consistently exhibit higher CSL, providing a strong signal of memorization. Complementary to this, Garg et al. Garg et al. [2024] and Ravikumar et al. Ravikumar et al. [2024] explore the curvature of the loss surface. Their results show that memorized examples lie in sharper regions of the landscape—i.e., with higher curvature. This often indicate less robust generalization and more brittle learning dynamics. Recent studies also demonstrate that memorization is not limited to supervised learning. Meehan et al. Meehan et al. [2023] uncover "déjà vu" memorization in self-supervised models, where training samples are memorized even without explicit labels. Kokhlikyan et al. Kokhlikyan et al. [2024] refine the measurement of this phenomenon, offering efficient evaluation tools for memorization in large SSL models. Similar memorization behavior is observed in vision-language models Jayaraman et al. [2024], where individual image or object information is retained by the model even beyond its intended abstraction level. **FocL offers an input-level simplification by suppressing background clutter entirely, reducing reliance on spurious correlations and shortcut cues Geirhos et al. [2020].** By restructuring the input itself, FocL shifts the learning task to focus on object-relevant features from the outset. Unlike techniques such as Mixup Zhang et al. [2018], CutMix Yun et al. [2019], or logit-adjustment methods, which alter training dynamics via label smoothing, augmentation, or reweighting, FocL tackles instance-level difficulty directly by improving input-label consistency through foveated, object-aligned supervision.

### Foveation, Robustness, and Efficient Learning

Foveation-inspired methods have been explored as mechanisms for improving robustness. R-Blur Shah et al. [2023] applies adaptive Gaussian blurring to simulate peripheral vision, improving resistance to adversarial attacks. Deza and Konkle Gant et al. [2021] use a Foveated Texture Transform to enhance both IID generalization and robustness. Active-vision systems Mukherjee et al. [2025] formulate a deep learning-based dorsal-ventral architecture by building on prior works such as FALcon Ibrayev et al. [2024b] and GFNet Wang et al. [2020], and demonstrate improved robustness in black-box transfer attack scenarios. By processing sequential glimpses at multiple fixation points, the approach enhances adversarial resilience for both CNNs and transformer-based ventral networks, particularly under natural and transferable adversarial inputs. Luo et al. Luo et al. [2015] apply CNNs to foveated regions, achieving strong robustness to perturbations. R-Warp Vuyyuru et al. [2020] and VOneBlock Dapello et al. [2020] embed cortical and retinal processing into CNNs. Harrington et al. Harrington and Deza [2022] show how robust models align with texture-based peripheral vision, and Shah et al. Shah et al. [2023] simulate peripheral degradation for robustness gains. **FocL introduces a simplified mechanism: a complete background cut-off via supervised crops. This restructuring results in cleaner, more learnable samples and exhibits a coupled effect; higher**

**adversarial perturbation energy required to flip predictions and lower Cumulative Sample Loss (CSL). Both serving as indicators of reduced memorization.**

FocL thus bridges perceptual inspiration with practical gains in generalization, memorization reduction, and efficient learning without requiring specialized architectures or costly training procedures.

## FocL glimpse generation algorithm details

**Overview.** The FOCL framework generates up to three object-centric glimpses per image, centered around a supervised fixation point derived from the annotated bounding box. These glimpses simulate small saccadic shifts near the object and reduce background clutter while preserving semantic alignment with the label. While the main paper outlines the high-level steps Figure 4, this section details the underlying algorithm and implementation used in our experiments.

**Step-by-step Procedure.** Given an annotated image $(x, y)$ with bounding box $b = (x_{\min}, y_{\min}, x_{\max}, y_{\max})$, we define the center $p \in \mathbb{R}^2$ of the box as the base fixation point. Glimpses are then constructed as follows:

- **Step 1: Sampling fixation candidates.** Around $p$, we sample up to $k_{\text{cand}}$ candidate centers $p_i$ using a uniform offset in both spatial directions. The maximum offset is set to a fraction $\alpha$ of the bounding box width/height, i.e.,

$$\Delta x, \Delta y \sim \mathcal{U}(-\alpha w, \alpha w), \quad \text{where } w = x_{\max} - x_{\min}.$$

  These jittered candidates simulate parafoveal fixations while remaining near the object center. Concretely, this defines a square jitter window around the center of the bounding box, within which candidate fixation points $p_i = p + (\Delta x, \Delta y)$ are sampled.

- **Step 2: Valid fixation selection.** For each candidate $p_i$, we compute a crop region whose aspect ratio and scale are randomly jittered using multiplicative factors $\beta_x, \beta_y \sim \mathcal{U}(1 - \beta, 1 + \beta)$. We retain up to $k \leq 3$ valid crops whose regions lie entirely within image bounds. This ensures all glimpses are valid, foreground-aligned views.

- **Step 3: Distortion-aware cropping.** For each selected $p_i$, the crop is resized to the model's input resolution. If the required resizing scale exceeds a threshold computed via an inverse crop ratio $\eta = 1/(1 - \texttt{max\_crop\_ratio})$. We first expand the crop window proportionally around its center (without crossing image bounds). This reduces geometric distortion when handling small or thin boxes.

- **Step 4: Aggregation.** Each image yields $k$ foveated crops $\{\text{Fov}_i(x, p_i)\}_{i=1}^k$. These are treated as label-consistent training samples and either randomly subsampled ($k = 1$) or stacked into a correlated mini-batch. Glimpses from the same image are never shuffled across batches, preserving the coherence of multi-view supervision.

**Implementation Notes.** The algorithm is implemented which exposes key parameters:

- `Offset_fraction` $= 0.2$: sets $\alpha$, the maximum offset for sampling.
- `Scale_jitter` $= 0.1$: sets $\beta$, the jitter range for scale and aspect ratio.
- `Max_crop_ratio` $= 0.2$: defines threshold $\eta$ to trigger crop expansion. The max crop ratio is a threshold parameter that controls how much a crop is allowed to be resized relative to the original bounding box before geometric distortion is considered too high.
- `Area_threshold` $= 0.2$: used to activate distortion-aware expansion for small objects.
- `Multi_crop` flag: if `True`, all $k$ glimpses are returned together; if `False`, one random crop is sampled per epoch.
- `Augmentation mode` entails {"conservative", "medium", "aggressive"}: scales the above hyperparameters accordingly.

This design ensures that glimpses maintain semantic alignment while providing spatial diversity around the object. The same framework supports single-glimpse ($k = 1$) and multi-glimpse ($k > 1$) supervision via a unified pipeline.

Table 5: FocL dataset and cropping hyperparameters.

| Parameter | Value |
|---|---|
| Offset fraction ($\alpha$) | 0.2 |
| Scale/aspect jitter ($\beta$) | 0.1 |
| Max crop ratio | 0.2 |
| Area threshold (for distortion-aware fallback) | 0.2 |
| Number of glimpses $k$ | 1 or 3 |
| Multi-crop batching | Enabled for $k > 1$ |
| Batch size | 128 (k=1), 64 (k=3) |
| Input resolution | $224 \times 224$ |
| Augmentation | **Medium** |

## Training Details and Reproducibility

**Dataset Preparation.** Following the setup in Meehan et al. [2023], we sample and curate our annotated dataset from ImageNet using the official codebase available at `https://github.com/facebookresearch/DejaVu`. All dataset checks, bounding box extraction, and curation pipelines were built on top of this repository. We adapt their utilities to generate the subset used for FocL, ensuring consistency in annotation quality and reproducibility of bounding box metadata.

We evaluate FocL across multiple ImageNet subsets with bounding box annotations. Our experiments use the following curated partitions:

- **Full-scale split.** Following Sections 4.1.1–4.1.3 of the main paper, we use the complete curated bounding box subset from ImageNet-1K (2012), comprising 482,187 images. This defines our **full-scale** setup. We apply a 94/6 train/validation split, yielding 453,254 training and 28,933 validation images. An additional held-out test set of 773 disjoint samples is used exclusively for evaluation. This split is used to study generalization (Section 4.1.1) and data-efficient learning (Section 4.3).

- **Controlled low-data splits.** We also define two disjoint 100K ImageNet subsets, referred to as **Partition A** and **Partition B**, each divided into an 85K/15K train-validation split. Partition A is used for most controlled analysis experiments:
  - The 85K train set from Partition A is used to analyze adversarial robustness (PGD distance), gradient norms, and memorization via cumulative sample loss (CSL).
  - The 15K validation set from Partition B is used for validation-time adversarial evaluation (Section 4.2).

Each image is preprocessed to extract either one or up to three foveated crops using the method described in Section 3.1. All crops are resized to $224 \times 224$ resolution. Inputs are normalized using the standard ImageNet mean and standard deviation.

**Model Architecture** We use a standard ResNet-50 He et al. [2015] architecture across all experiments, with no architectural differences between Standard and FocL models.

**Training Configuration.** All models are trained for 90 epochs using SGD with momentum 0.9 and weight decay $1 \times 10^{-4}$. The initial learning rate is set to 0.1 and decayed by a factor of 0.1 every 30 epochs. We use a batch size of 64 *per worker*, which is flattened across multiple glimpses during multi-crop training (e.g., $k = 3$ glimpses per image). We use a batch size of 128 for $k = 1$ glimpse per image. Each training sample is augmented using standard ImageNet transforms: random resized crop, horizontal flip, and color jitter. All experiments are tracked using Weights & Biases.

**Optimization and Learning Rate Schedule.** We use the standard cross-entropy loss as the training objective. Optimization is performed using stochastic gradient descent (SGD) with momentum set to 0.9 and weight decay of $1 \times 10^{-4}$. The initial learning rate is 0.1, decayed by a factor of 0.1 every 30 epochs using a StepLR scheduler. All models are trained with mixed precision using PyTorch's `GradScaler` for improved stability and efficiency.

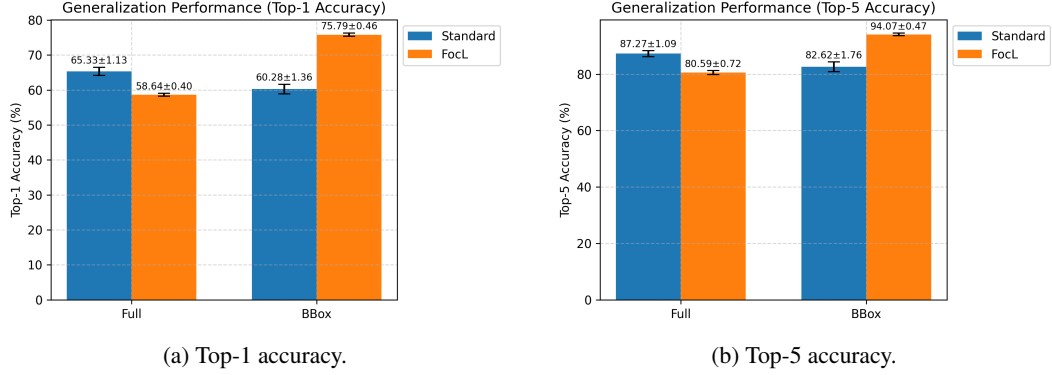

(a) Top-1 accuracy.  (b) Top-5 accuracy.

Figure 9: Generalization performance under full-image and bounding box inference. FocL consistently improves BBox performance while maintaining reasonable full-image performance. Error bars denote standard deviation over 3 runs.

**Reproducibility and Statistical Significance**   We ensure statistical rigor by repeating key experiments across multiple random seeds and reporting mean and standard deviation where applicable:

- **Generalization experiments (Section 4.1.1):** All models evaluated using both oracle bounding box inference and FALcon inference are trained across 3 random seeds. We report the mean Top-1 accuracy in the main paper, and include standard deviation as error bars in the Supplementary.

- **Data-efficient learning (Section 4.3):** To assess consistency in low-data settings, we train models on Partition A (100K subset) across 5 different random seeds. Aggregate results with error bars are presented in the Supplementary.

- **CSL and adversarial robustness (Section 4.2):** Cumulative sample loss is computed by logging per-sample training loss across all 90 epochs on the 100K subset. We also evaluate PGD-based adversarial distance across 5 different $\ell_2$ budgets ($\epsilon$) on the same partition.

**Compute and Environment.**   All models are trained on NVIDIA A40 GPUs with 48GB memory per device. We follow the same training hyperparameters and optimization settings for both Standard and FocL models. The full training pipeline, configuration scripts, and an environment file are included in the code submission.

*An environment file named requirements.txt is included in the supplementary materials to ensure full reproducibility.*

## Generalization Results

**Generalization Performance.**   Figure 9 reports the Top-1 and Top-5 accuracy for models trained on 482K samples and evaluated on the held-out 773-image test set under both full-image and bounding box inference. Mean accuracy and standard deviation are shown as error bars over 3 random seeds. FocL models exhibit substantial improvements in the BBox setting, confirming their object-centric learning advantage. **FocL also achieves lower standard deviation on the 773-image held-out test set, indicating more stable generalization and reduced sensitivity to initialization or data order.**

**Evaluation with FALcon.**   In Figure 10, we compare the evaluation performance across five metrics using FALcon inference. Again, we report mean and standard deviation over 3 seeds. Top-1 is accuracy on full images without any glimpse localization. **Notably, FocL models consistently show lower standard deviation across metrics compared to their standard counterparts. This suggests more stable training and improved reliability in capturing label-relevant structure, likely due to the reduced influence of background noise and more focused gradient updates.**

**Ablation: Role of Glimpse Diversity and Distortion-Aware Cropping.**   We evaluate three variants of our approach to isolate the impact of foveation design. The **vanilla crop** baseline resizes a single

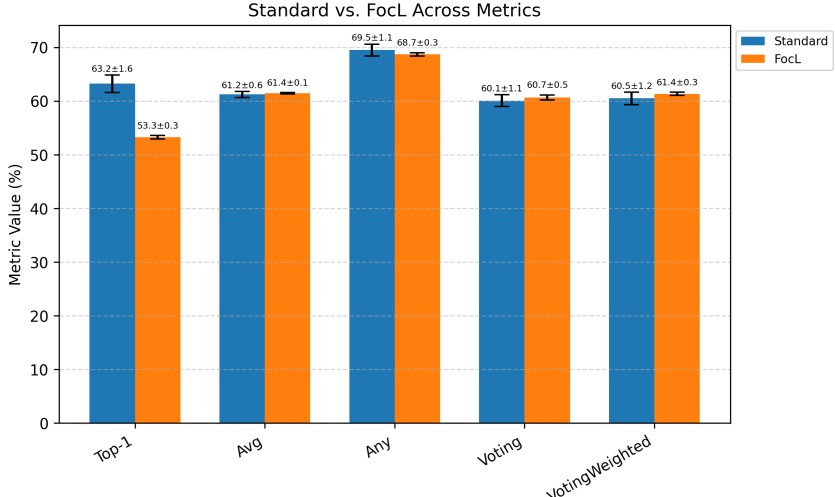

Figure 10: Comparison of Standard vs. FocL across clean evaluation metrics under FALcon inference. FocL achieves comparable or higher performance on most metrics with significantly lower variance.

Table 6: Oracle bounding box inference (773 samples). Top-1 and Top-5 accuracy across three FocL variants: vanilla crop (no distortion or saccadic shift), distortion-aware single-crop (SC), and multi-crop (MC).

| Training | Top-1 | Top-5 |
|---|---|---|
| FocL Vanilla Crop | 74.39 | 92.01 |
| FocL MultiGlimpseDA SC | 74.51 | 92.88 |
| FocL MultiGlimpseDA MC | **75.79** | **94.07** |

bounding box crop without any distortion-aware expansion or parafoveal variation. The **FocL Single-Crop (SC)** variant applies scale-jittered cropping, reducing overfitting to tight box boundaries. Finally, the full **FocL Multi-Crop (MC)** model adds viewpoint diversity via multiple glimpses from jittered fixation points. As shown in Tables 6 and 7, both oracle and FALcon evaluations improve progressively from vanilla to SC to MC. These results highlight the complementary benefits of spatial variation and distortion-aware logic in object-centric learning.

## Adversarial Robustness

**Setup**  We evaluate adversarial resistance by computing the minimum $\ell_2$ perturbation required to flip model predictions using PGD attacks Madry et al. [2018]. All experiments are conducted on a balanced ImageNet subset with 100 samples per class and an 85/15 train-validation split. We use a PGD-$\ell_2$ attack with 10 steps, random initialization, and random restarts enabled. The step size is set to $\alpha = \epsilon/10$, and we sweep the perturbation budget $\epsilon \in \{0.0, 0.25, 0.5, 0.75, 1.0\}$.

To construct a clean and balanced evaluation protocol, we select 15,000 correctly predicted samples from the training set (Partition A) and 15,000 correctly predicted samples Partition B. This forms the validation set results for Partition A (unseen). This ensures that the evaluation is based on semantically aligned, clean samples and keeps the number of inputs consistent across training and validation settings. We compute both the robustness curves and the mean adversarial distance on these subsets, allowing for a fair comparison between FocL and standard models.

**Mean Adversarial Distance**  To quantify robustness, we compute the average adversarial distance:

$$\bar{d} = \frac{1}{N} \sum_{i=1}^{N} \|\delta_i\|_2 \quad \text{where} \quad f(x_i + \delta_i) \neq y_i$$

Table 7: FALcon inference (class-agnostic). Evaluation of generalization performance across five voting-based metrics.

| Training | Top-1 | Avg | Any | Voting | Voting-Wtd |
|---|---|---|---|---|---|
| FocL Vanilla Crop | 43.05 | 55.60 | 62.90 | 54.50 | 55.35 |
| FocL MultiGlimpseDA SC | 50.25 | 60.30 | 68.10 | 59.50 | 60.05 |
| FocL MultiGlimpseDA MC | **53.27** | **61.45** | **68.72** | **60.68** | **61.37** |

Here, $\delta_i$ denotes the smallest perturbation (in $\ell_2$ norm) found via PGD that causes a misclassification. We find that the standard model has $\bar{d} = 0.3806$, while FocL achieves $\bar{d} = 0.6169$—a 62% increase. This gap reflects a substantial improvement in robustness. We have mentioned this in the main manuscript. Higher adversarial distance implies that more energy is required to change the model's decision, suggesting a more stable and semantically aligned representation. These results support the argument that standard models overfit to incidental background cues, while FocL focuses learning on foreground-relevant features that are inherently harder to perturb.

## Cumulative Sample Loss (CSL) as a proxy for learning difficulty

**Setup** We evaluate cumulative sample loss (CSL) as a proxy for sample difficulty and memorization. The setup follows the same 85K/15K train-validation split used in the robustness analysis. CSL quantifies how difficult a sample is to learn by accumulating its training loss over epochs. Formally, for a training sample $z = (x, y)$, the cumulative sample loss over $T$ epochs is defined as:

$$\text{CSL}(z) = \sum_{t=1}^{T} \mathcal{L}(f_{\theta_t}, z)$$

where $\mathcal{L}$ denotes the cross-entropy loss, $f_{\theta_t}$ is the model at epoch $t$, and $z$ is the training sample. For the FocL model, the sample is represented as $\text{Fov}(x, y)$, denoting a foveated crop centered on the object. High CSL values correspond to samples that remain difficult across multiple epochs and are more likely to be memorized rather than learned robustly. For fair comparison, we evaluate CSL for the FocL single-crop variant to match the standard model's single-view training.

In Figure 11, we provide a per-sample analysis demonstrating how FocL facilitates easier learning. For each class (Llama, Peacock, Beaver), the green-boxed examples (FocL) consistently show lower cumulative sample loss (CSL) compared to their red-boxed full-image counterparts. This shift in CSL values explains the leftward shift in the aggregate CSL distribution observed in the main paper, supporting our claim that FocL improves learning stability and efficiency.

## Gradient Norm Analysis

**Setup** To probe the optimization dynamics of FocL, we analyze the magnitude of gradients during training. Specifically, we compute the $\ell_2$ norm of gradients with respect to all model weights on the training set of Partition A (85K samples from the 100K ImageNet subset). Gradient norms are logged throughout training for both the standard model and the FocL single-crop variant. This analysis provides insight into training stability and the ease of optimization under different input regimes.

FocL exhibits consistently smaller gradient magnitudes compared to standard training, suggesting a smoother optimization landscape. The standard model records a mean gradient norm of $3.81 \times 10^4$ with a standard deviation of $2.26 \times 10^4$, while FocL reports a lower mean of $2.07 \times 10^4$ and a standard deviation of $1.28 \times 10^4$. When normalized by the total number of ResNet-50 parameters ($\sim 2.56 \times 10^7$), the per-parameter gradient norm drops from $1.49 \times 10^{-3}$ (standard) to $8.08 \times 10^{-4}$ (FocL)—a relative reduction of approximately 45.8%. This substantial drop suggests that FocL's object-centric inputs result in less gradient noise and more stable optimization, aligning with our findings on faster convergence and lower memorization.

Table 8: Evaluation in the low-data regime using the 50K test set from Partition B. With 41.18% fewer training samples (50 vs. 85 per class), the FocL Single Crop model achieves comparable or better performance than the standard model trained on the full set. All results report mean ± standard deviation across 5 random data partitions.

| Tested on | Dataset Size (K) | Top-1 | Top-5 |
|---|---|---|---|
| Full Image (Standard) | 85 | 44.30 | 68.56 |
| Full Image (Standard) | 50 | $26.91 \pm 1.20$ | $49.36 \pm 1.17$ |
| Bounding Box (FocL SC) | 50 | $45.04 \pm 0.93$ | $70.30 \pm 0.85$ |

## Data efficiency

**Setup for Low-Data Regime**    To evaluate data efficiency, we train all models on Partition A of the 100K balanced ImageNet subset. The standard baseline uses 85 training samples per class, while low-data models are trained with 50 samples per class. These 50-per-class subsets are derived from five random data partitions of Partition A (i.e., five distinct data seeds). All models are trained for 90 epochs using SGD with momentum and a step learning rate scheduler (decay at epochs 30 and 60), with otherwise identical hyperparameters.

Evaluation is performed on a fixed 50K test set from Partition B. The standard model is evaluated on full-resolution images, while FocL models are evaluated using bounding box–aligned crops. As shown in Section 4.1.1 of the main paper, full-image models underperform when evaluated on oracle bounding boxes. Therefore, we report results using their respective optimal evaluation inputs. For FocL, we use the single-crop variant, consistent with the setup in Section 4.3.

**Analysis.**    FocL demonstrates superior data efficiency in the low-data regime as well. With only 50 training samples per class (41.18% fewer than the standard baseline), it achieves a **Top-1 accuracy of 45.02%** and **Top-5 of 70.30%**, outperforming the standard model trained on 85 samples/class (Top-1: 44.30%, Top-5: 68.56%). This margin holds consistently across 5 random data partitions. **These results are statistically consistent across data splits, highlighting FocL's robust ability to leverage object-centric signals even in lower data regime.**

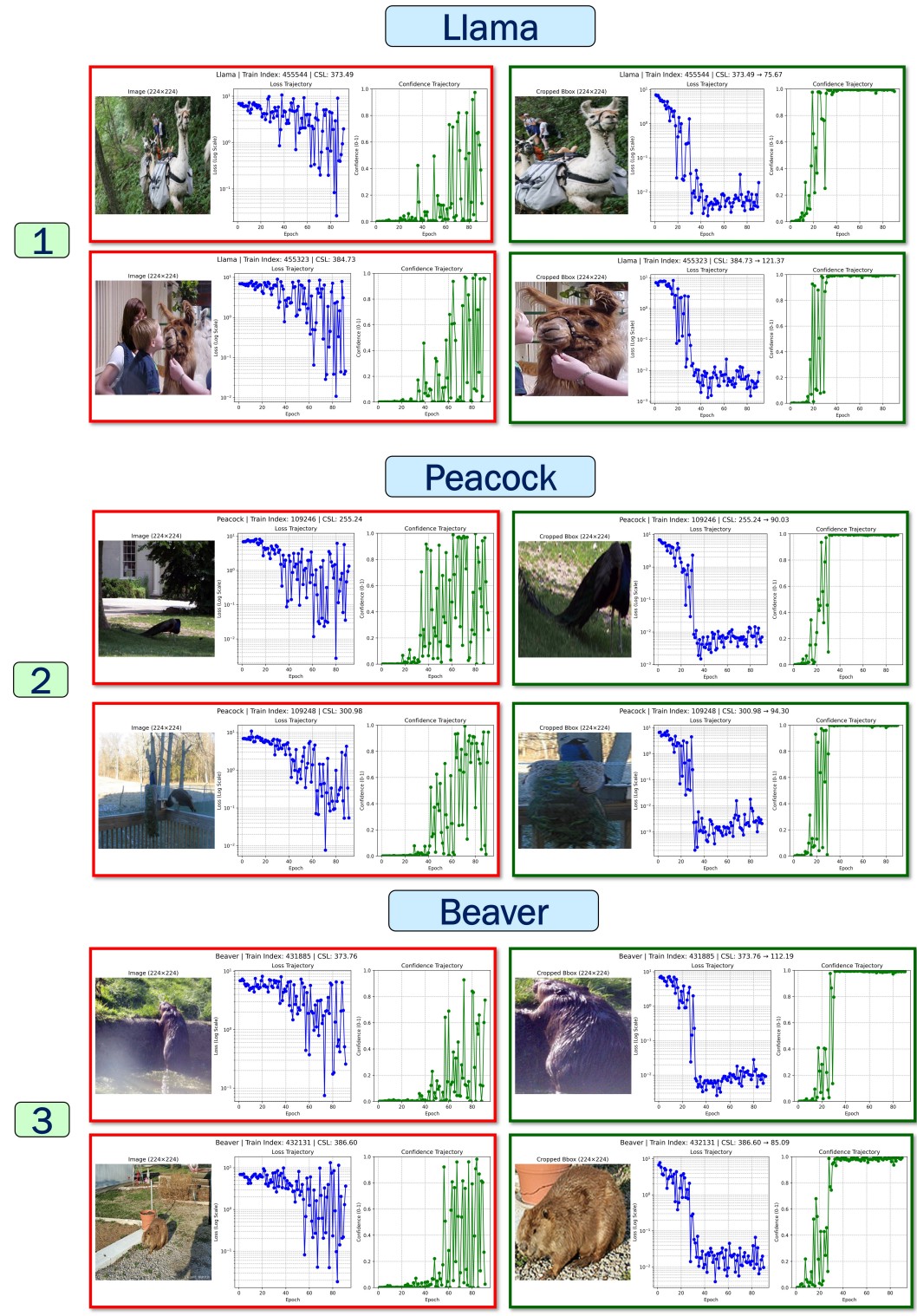

Figure 11: Sample visualization of CSL dynamics across three classes—**Llama**, **Peacock**, and **Beaver**. Each row compares full-image (left, red box) vs. FocL-based cropped inputs (right, green box). Across classes, FocL leads to faster convergence (loss trajectory), more confident predictions (confidence trajectory), and substantially lower cumulative sample loss (CSL). These patterns are consistent with aggregate statistics shown in CSL distributions, train-validation loss curves, and gradient norm plots.

