# OpenReview forum: "From Clutter to Clarity: Visual Recognition through Foveated Object-Centric Learning (FocL)"
_NeurIPS.cc/2025/Workshop/Reliable_ML — NeurIPS 2025 - Reliable ML Workshop_

### Official Review · Reviewer_3XAz · 2025-09-24

**Rating:** 6
**Confidence:** 4

**Review:**

The paper introduces FocL (Foveated Object-Centric Learning), a biologically inspired training strategy for visual recognition. Instead of training on full images that often contain clutter and spurious correlations, FocL replaces them with object-centric crops (foveated glimpses) generated by jittering bounding-box centers to simulate saccades. Experiments on ImageNet show that FocL improves generalization on oracle crops by up to 15% and boosts out-of-distribution accuracy from ImageNetV1 to V2 by more than 7 percentage points when paired with SAM. FocL also enhances robustness: adversarial perturbation budgets required to flip predictions increase by 61%, and the top 1% most memorized samples see a 62.5% reduction in cumulative sample loss. The approach converges faster and achieves strong performance with 56% less data, suggesting improvements in both reliability and efficiency.

Strengths:
Innovative use of foveation and saccadic simulation, grounding the method in human vision mechanisms.
Significant gains in out-of-distribution generalization, adversarial robustness, and sample efficiency.

Weaknesses:
Most experiments are on ImageNet. Testing on a diverse range of datasets could strengthen the findings.